# iFlame: Interleaving Full and Linear Attention for Efficient Mesh Generation

## Abstract

This paper proposes iFlame, a novel transformer-based network architecture for mesh generation. While attention-based models have demonstrated remarkable performance in mesh generation, their quadratic computational complexity limits scalability, particularly for high-resolution 3D data. Conversely, linear attention mechanisms offer lower computational costs but often struggle to capture long-range dependencies, resulting in suboptimal outcomes. To address this trade-off, we propose an interleaving autoregressive mesh generation framework that combines the efficiency of linear attention with the expressive power of full attention mechanisms. To further enhance efficiency and leverage the inherent structure of mesh representations, we integrate this interleaving approach into an hourglass architecture, which significantly boosts efficiency. Our approach reduces training time while achieving performance comparable to pure attention-based models. To improve inference efficiency, we implemented a caching algorithm that almost doubles the speed and reduces the KV cache size by seven-eighths compared to the original Transformer. We evaluate our framework on ShapeNet and Objaverse, demonstrating its ability to generate high-quality 3D meshes efficiently. Our results indicate that the proposed interleaving framework effectively balances computational efficiency and generative performance, making it a practical solution for mesh generation. The training takes only 2 days with 4 GPUs on 39k meshes with a maximum of 4k faces on Objaverse.

## 1 Introduction

3D content generation is crucial for various domains, including virtual reality, gaming, industrial design, and digital content creation. The ability to automatically generate high-quality 3D models is fundamental to these applications, enabling more immersive experiences and efficient design workflows.

3D objects can be represented in various formats, including point clouds, voxels, implicit functions, and meshes. Among these representations, triangular meshes stand out for their widespread adoption in graphics pipelines, efficient rendering capabilities, and ability to represent complex geometries with sharp features. However, generating high-quality meshes remains challenging due to their irregular structure and the requirement to maintain geometric and topological consistency.

Recent advances in deep learning have led to significant progress in mesh generation (Weng et al., 2024b; Wang et al., 2025; 2024; Chen et al., 2024a;a; Tang et al., 2024b; Hao et al., 2024). Most existing approaches focus on conditional generation, where meshes are created based on point clouds or reference images. While these methods can handle complex geometries, they often rely on compressed mesh representations or intermediate formats, which may not fully preserve the original mesh properties. In contrast, unconditional mesh generation, which aims to learn the underlying distribution of 3D shapes without additional inputs, has been limited to relatively simple geometries (around 800 faces) as demonstrated in works like MeshGPT (Siddiqui et al., 2024) and MeshXL (Chen et al., 2025).

Current mesh generation network architectures rely heavily on attention layers (Vaswani et al., 2017), which have shown remarkable capability in capturing complex geometric relationships. However, the quadratic computational complexity of full attention with respect to sequence length poses significant scalability challenges, especially for high-resolution meshes. This limitation becomes

particularly apparent when generating meshes with thousands of faces, where computational resources become a bottleneck.

Recently, there has been more interest in linear attention (Katharopoulos et al., 2020) for sequence modeling tasks, offering linear computational complexity but typically achieving lower performance compared to attention-based architectures (Qin et al., 2022a). This presents an interesting trade-off between computational efficiency and model expressiveness.

To address this trade-off, we propose an interleaving autoregressive mesh generation framework that combines the efficiency of linear attention (Qin et al., 2024b) with the expressive power of full attention mechanisms (Vaswani et al., 2017). By further integrating our interleaving approach into an hourglass architecture (Hao et al., 2024; Nawrot et al., 2021), we achieve even greater resource efficiency. The hourglass structure enables multi-scale processing through systematic downsampling and upsampling operations, effectively capturing information at various representational scales critical for coherent mesh generation. This hierarchical design processes information at coordinate scale, vertex scale, and face scale simultaneously, allowing the model to reason about mesh geometry at different levels of abstraction. The ability to handle these interconnected spatial representations is particularly well-suited for 3D meshes, where relationships between vertices and faces must be modeled consistently across multiple scales. We evaluate our method on ShapeNet (Chang et al., 2015) and Objaverse (Deitke et al., 2023), demonstrating its ability to generate high-quality 3D meshes efficiently.

As shown in Fig. 1, our model maintains comparable performance to pure attention-based models while significantly improving computational efficiency across multiple dimensions. During training of our ShapeNet (Chang et al., 2015) experiments 800 faces), our approach reduces training time requirements by **46%** and memory consumption by **38%** compared to full attention-based architectures. These efficiencies continue during inference, achieving similar token accuracy and perplexity while increasing throughput by **82%** (81.9 tokens/second vs. 45 tokens/second). Our strategy further enhances efficiency by reducing cache memory requirements by **88%** (0.8GB vs. 6.6GB) compared to the standard transformer.

Thanks to the efficient Linear attention mechanisms strategically incorporated into our design, our interleaved hourglass architecture demonstrates increasingly favorable computational characteristics as sequence length grows. This scaling efficiency enables the generation of meshes with up to 4,000 faces using limited computational resources (4 A100 GPUs)—a significant advancement over existing unconditional generation methods (also see the table for training GPU days on Objaverse).

|  | GPU Days | # Faces |
|---|---|---|
| M.XL (Chen et al., 2025) | 968 | 800 |
| M.Any (Chen et al., 2024a) | 32 | 800 |
| M.Any v2 (Chen et al., 2024b) | 96 | 1600 |
| E.R. (Tang et al., 2024b) | 560 | 4000 |
| Ours | 8 | 4000 |

Our comprehensive results indicate that the proposed **iFlame** framework effectively balances computational efficiency and generative performance, making it a practical solution for mesh generation tasks.

The main contributions of this work are:

- A novel interleaved hourglass architecture that strategically combines full and linear attention mechanisms, achieving high-quality mesh generation while maintaining accuracy comparable to pure attention models

- Significant efficiency gains in inference speed (1.8× faster) and cache usage (88% reduction)

- Successful scaling of unconditional mesh generation to substantially higher resolution (up to 4,000 faces) than previously possible with comparable computational resources

## 2 RELATED WORK

### 2.1 3D SHAPE REPRESENTATION AND GENERATION

Various representations have been developed for 3D shape generation, each with distinct advantages and limitations. Point cloud-based methods (Fan et al., 2017; Achlioptas et al., 2018; Zeng et al.,

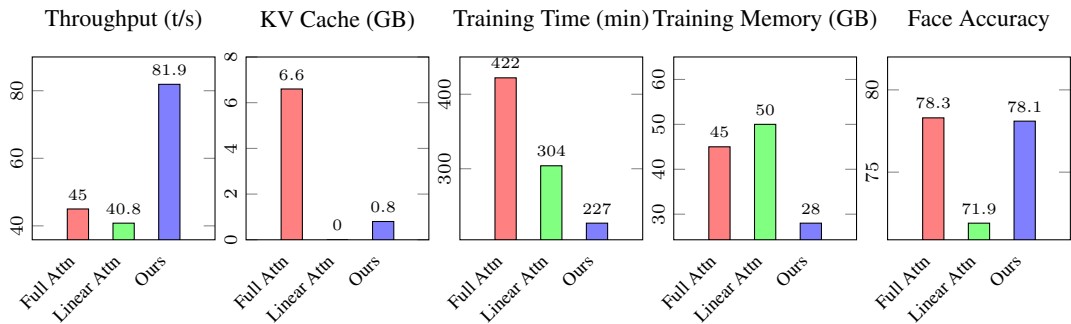

Figure 1: Performance comparison of our iFlame architecture. **(a)** Our model achieves 1.8× higher inference throughput (81.9 t/s vs. 45.0 t/s). **(b)** Our model maintains low KV cache usage (0.8GB) while full attention requires 8.3× more memory when generating 4000 faces. **(c, d, e)** Our model reduces training time by 46% (227 min vs. 422 min), requires 38% less GPU memory during training (28GB vs. 45GB per GPU), and maintains face accuracy (78.1% vs. 78.3%) compared to baseline methods on ShapeNet with 2B tokens.

2022; Cheng et al., 2022; Zhou et al., 2021; Luo & Hu, 2021) represent shapes as unordered sets of 3D points, offering simplicity but often lacking surface connectivity. Voxel-based approaches (Wu et al., 2015; Choy et al., 2016; Xie et al., 2020; Mittal et al., 2022) discretize 3D space into regular grids, providing explicit volumetric representations at the cost of resolution limitations due to memory constraints.

Implicit function representations such as Signed Distance Functions (SDFs) (Park et al., 2019; Chibane et al., 2020; Zhang & Wonka, 2024a; Zheng et al., 2022) and occupancy networks (Mescheder et al., 2019; Zhang et al., 2023; Zhang & Wonka, 2024b) encode shapes as continuous functions, enabling high-resolution surface extraction but requiring post-processing to obtain explicit geometry.

The field of autoregressive mesh generation has seen significant advancement in recent years. Poly-Gen (Nash et al., 2020) pioneered the approach of separately generating points and faces in an autoregressive manner to construct 3D meshes. MeshGPT (Siddiqui et al., 2024) marked a breakthrough by introducing a unified token-based autoregressive framework for mesh generation, achieving impressive results. This was followed by Mesh Anything (Chen et al., 2024a;b), which extended the paradigm to conditional mesh generation.

MeshXL (Chen et al., 2025) demonstrated the scalability of autoregressive mesh generation on the large-scale Objaverse dataset (Deitke et al., 2023). Llama-Mesh (Wang et al., 2024) successfully explored unifying 3D mesh generation with language models, bridging the gap between NLP and geometric modeling. Research on improving token efficiency has progressed with PivotMesh (Weng et al., 2024a), which explored token length compression techniques. EdgeRunner (Tang et al., 2024b) employed classical mesh processing algorithms to further compress sequence length, while BPT (Weng et al., 2024b) and Nautilus (Wang et al., 2025) optimized mesh point indexing methods to achieve better compression ratios. MeshTron (Hao et al., 2024) leveraged an hourglass transformer architecture to improve efficiency, demonstrating excellent results for point cloud to mesh conversion. While most prior work has focused on conditional generation or sequence compression techniques, the area of high-resolution unconditional mesh generation remains relatively unexplored. To date, only MeshGPT (Siddiqui et al., 2024) and MeshXL (Chen et al., 2025) have demonstrated the ability to generate meshes with approximately 800 faces without conditioning. This represents an important research direction worthy of further exploration. Our work advances the state-of-the-art by increasing this limit to 4,000 faces.

## 2.2 LINEAR AND HYBRID ATTENTION MECHANISMS

Transformer architectures have revolutionized sequence modeling across domains, but their quadratic computational complexity with respect to sequence length creates significant scaling challenges. This limitation has catalyzed extensive research into more computationally efficient alternatives that preserve modeling capabilities.

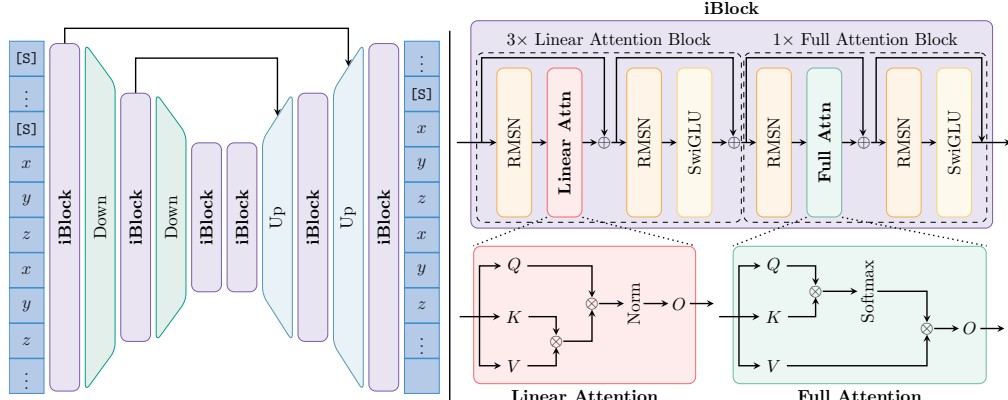

Figure 2: The architecture of our iFlame

Linear attention mechanisms fundamentally reformulate the attention operation to achieve linear complexity with sequence length. Performer (Choromanski et al., 2021) extended this approach by approximating the softmax kernel using random feature projections, while Cosformer (Qin et al., 2022b) introduced a position-aware cosine similarity-based attention mechanism. Despite these innovations, linear attention models often struggle with the expressivity-efficiency trade-off (Qin et al., 2022a).

An alternative approach leverages recurrent architectures with enhanced expressivity. Mamba (Gu & Dao, 2023) introduced selective state space modeling (S4) for efficient sequence processing, achieving linear complexity while maintaining competitive performance. Its successor, Mamba2 (Dao & Gu, 2024), incorporates a gated update mechanism that further improves performance on language modeling and long-context understanding tasks. Similarly, RWKV (Peng et al., 2023) combines RNN-like computation with transformer-like expressivity to achieve efficient inference.

More sophisticated hybrid designs include Griffin (De et al., 2024), which interleaves gated linear attention with local attention patterns, and Samba (Ren et al., 2024), which integrates state space models with standard attention layers. DeltaNet (Yang et al., 2024) employs selective memory updates based on the delta rule, demonstrating particularly strong results for in-context retrieval tasks.

Minimax-text-1 (Li et al., 2025) pioneered the implementation of an interleaving attention structure in large language models, demonstrating the scalability and effectiveness of this architectural approach. Our work further enhances the efficiency of the interleaving strategy. While many recent approaches have introduced complex gating mechanisms or specialized activation functions to improve performance, our design philosophy emphasizes architectural simplicity and computational efficiency.

## 3 METHOD

We first describe our mesh representation, then recall the formulation of attention and linear attention before introducing our proposed iFlame architecture.

### 3.1 MESH REPRESENTATION

We conceptualize mesh generation as an autoregressive sequence generation problem, like MeshGPT, Meshtron and others. A triangular mesh $\mathcal{M} = (\mathcal{V}, \mathcal{F})$ has vertices $\mathcal{V} = \{\mathbf{v}^i \in \mathbb{R}^3\}_{i=1,...,M}$ and faces $\mathcal{F} \subset \mathcal{V} \times \mathcal{V} \times \mathcal{V}$. The number of vertices is $|\mathcal{V}| = M$ and the number of faces is $|\mathcal{F}| = N$.

To facilitate autoregressive generation, a commonly used approach (*e.g.*, MeshGPT (Siddiqui et al., 2024)) is to flatten mesh vertices $\mathcal{V}$ into an ordered sequence by adopting a consistent convention. For example, the sequence is ordered by its $z$-coordinate, then by $y$-coordinate, and finally by $x$-coordinate, all in ascending order. We still use the symbol $\mathcal{M}$ to denote the flattened sequence to

simplify notation. The sequence is also augmented with special tokens: start token [S], end token [E], and padding token [P].

To allow autoregressive sampling, vertex coordinates are quantized into $b$ bins, balancing geometric precision and computational efficiency. We use $b = 128$ for a fair comparison with MeshGPT, but we also experimented with up to 1024 bins without much performance penalty. This representation allows us to model the mesh generation process as: $p(\mathcal{M}) = \prod_t p(q_t \mid \mathbf{q}_{<t})$ where $q_t$ denotes each quantized coordinate value in the sequence and $t$ means the position in the sequence.

However, this serialization approach faces significant challenges when scaling to complex meshes. When flattened into a sequence, a mesh with $N$ faces results in a sequence of length $9N$ (3 vertices per face, each with 3 coordinates). Standard transformer (with full attention) architectures with context lengths of 4096 or 8192 tokens can only process a limited number of faces. Existing methods struggle to scale efficiently due to the quadratic complexity of attention mechanisms, which are both time and memory-intensive. While Meshtron attempts to address this by using truncated sequences during training, experiments from BPT (Weng et al., 2024b) show that this approach often leads to discontinuities in the generated meshes.

To address these scaling challenges, we first revisit the fundamental attention mechanisms that underlie transformer architectures. Full attention offers strong expressiveness but lacks efficiency, while linear attention provides efficiency at the cost of effectiveness. This insight motivates our interleaving approach.

## 3.2 PRELIMINARY

**Attention.** The original transformer design (Vaswani et al., 2017) utilizes a self-attention mechanism that can be mathematically represented as:

$$\text{Softmax}\left(QK^\top/\sqrt{d}\right)V \tag{1}$$

In this formulation, $Q$, $K$, and $V \in \mathbb{R}^{n \times d}$ represent the query, key, and value matrices, respectively, where $n$ refers to the sequence length and $d$ indicates the feature dimensionality. The computational burden of this approach arises from calculating the full attention matrix $QK^\top \in \mathbb{R}^{n \times n}$, resulting in $O(n^2 d)$ complexity during training. Flash Attention (Dao et al., 2022; Dao, 2023a) significantly improves the efficiency of computing this operation through careful memory management, without changing the mathematical formulation. For autoregressive generation tasks, when producing the $m$-th token, the model requires $O(md^2)$ computations as it must attend to all previously generated tokens.

**Linear Attention.** As an alternative, linear attention variants like Lightning attention (Qin et al., 2024a) reformulate the attention mechanism by removing the computationally intensive softmax and scaling operations. This approach can be expressed as:

$$\text{Norm}((QK^\top)V) \tag{2}$$

To achieve better computational efficiency, this expression can be algebraically rearranged into its equivalent linear form:

$$\text{Norm}(Q(K^\top V)) \tag{3}$$

This rearrangement significantly improves efficiency by reducing the computational complexity to $O(nd^2)$ during the training phase, with benefits becoming more pronounced as sequence length increases. During inference, linear attention maintains a consistent $O(d^2)$ computational complexity regardless of context length by progressively updating the $K^\top V$ term, enabling efficient processing of sequences of any practical length.

## 3.3 INTERLEAVING ATTENTION BLOCK

After analyzing the trade-offs between full and linear attention, it is natural to conjecture that an interleaving approach can significantly reduce computational costs while maintaining model expressiveness. This insight led us to design an interleaving block architecture that strategically combines both attention mechanisms.

Minimax-text-1 (Li et al., 2025) was the first large language model to implement an interleaving attention structure. However, like other modern linear attention implementations, their approach often employs KQV SiLU and gate mechanisms, which consume substantial computational resources. To enhance efficiency and maintain structural symmetry with full attention, we adopt a minimalist linear attention design, referred to as **simplified linear attention**, as illustrated in our pipeline (Fig. 2).

Each transformer block consists of a self-attention layer followed by a feed-forward network (also called MLP or fully connected neural network), both equipped with residual connections and pre-normalization layers. Given an input sequence $X \in \mathbb{R}^{n \times d}$, the block computations are formulated as:

$$X \leftarrow X + f(\text{RMSNorm}(X))$$
$$Y \leftarrow X + \text{SwiGLU}(\text{RMSNorm}(X))$$

(4)

The attention function $f$ can adaptively switch between the full attention mechanism in Equation equation 1 and the linear attention variant in Equation equation 3. For causal attention, appropriate masking is applied to ensure that each position can only attend to previous positions. Additionally, Rotary Position Embeddings (RoPE (Su et al., 2021)) are incorporated into the computation of $Q$ and $K$ to encode positional information.

### 3.4 HOURGLASS STRUCTURE

Building upon our interleaving attention blocks, we further enhance efficiency by leveraging the inherent hierarchical structure of meshes. The natural face-vertex-coordinate structure of meshes is particularly well-suited for an hourglass network architecture like Meshtron, which can substantially reduce computational demands.

The Hourglass structure operates across three scales through six interconnected blocks. The first block encodes coordinates to extract fine-grained features ($\Phi_{\text{coord}}$), which are downsampled to vertex features for the second block. This vertex encoder returns vertex feature ($\Phi_{\text{vertex}}$), further downsamples to produce face features for the network core (blocks three and four). To maintain causality, face features, which include all vertex data, cannot be directly fused with the vertices in a face. Instead, a shifting mechanism is used; for example, with three vertices defining a face, a shift of 2 ensures that only the last vertex receives complete face information. The face features are combined with shifted vertex features ($\Phi_{\text{vertex}}$) from the second block and processed through the fifth block. After upsampling to $\Psi_{\text{coord}}$, these features are integrated with the shifted coordinate features ($\Phi_{\text{coord}}$) and processed by the final block to produce the output. This multi-scale architecture enables efficient feature extraction and integration, enhancing mesh processing capabilities.

### 3.5 INFERENCE

The hourglass architecture naturally processes information at multiple scales through its downsampling and upsampling operations. Implementing this hierarchical approach for efficient inference requires careful design to manage the complex token dependencies and state tracking across different resolution levels.

We designed an inference algorithm that respects these architectural constraints while minimizing unnecessary computation. In our method, the base encoder and final decoder process every token, the intermediate layers process every third token, and the bottleneck layers handle every ninth token. This hierarchical approach necessitates maintaining circular buffers to effectively track states across different resolution levels.

The key efficiency of our approach comes from the strategic integration of three linear attention layers and one full attention layer in our **iBlock**. The linear layers require an $O(1)$ key-value cache, while full attention requires $O(n)$, enabling a 75% reduction in cache memory usage. Furthermore, the hourglass structure further decreases the cache consumption, resulting in an overall reduction of over 87% compared to full attention. Our implementation leverages a multi-resolution processing pipeline and the low cache property of the **iBlock**, allows the system to capture dependencies at various temporal scales while maintaining manageable computational and memory requirements.

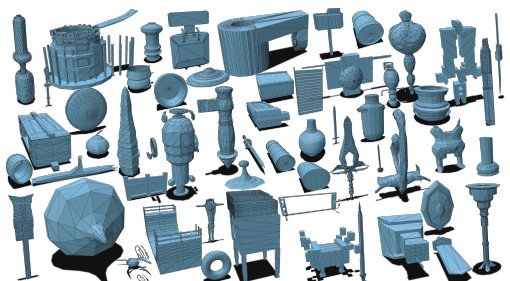

Figure 3: Comparison of mesh generation quality between MeshGPT and iFlame on ShapeNet.

Figure 4: Generative results on Objaverse.

## 4 EXPERIMENTAL SETUP

### 4.1 IMPLEMENTATION DETAILS

For ShapeNet, we train a 24-layer transformer (76M parameters) with embedding dimension 512 and 16 attention heads. We use 4 A100 GPUs with batch size 64, Muon (Jordan et al., 2024; Liu et al., 2025) optimizer with $1 \times 10^{-3}$ learning rate, 2-epoch warmup and cosine decay. For Objaverse, we scale to a 24-layer transformer (0.3B parameters) with embedding dimension 1024 and 16 attention heads. Training uses 4 A100 GPUs with batch size 16, for 2 days. Muon optimizer with $3 \times 10^{-4}$ learning rate, following the same warmup and decay schedule. To optimize memory usage and computational efficiency, we employ Flash Attention 2 (Dao, 2023b) for full attention computation and Lightning Attention (Qin et al., 2024a) for linear attention operations.

We set up the baseline as follows: full attention, linear attention, interleaving full and linear attention (**I**), interleaving full and simplified linear attention (**I+S**), interleaving full and simplified linear attention with hourglass architecture (**I+S+H**). For inference, we employ nucleus sampling with top-p value of 0.95 and top-k value of 50, which provides a good balance between diversity and quality in the generated meshes.

### 4.2 DATA PROCESSING AND AUGMENTATION

Following the MeshGPT preprocessing pipeline, we filter the ShapeNet dataset to include meshes with fewer than 800 faces, resulting in 28,570 meshes for training and 430 for testing. We utilize the training subset provided by LGM (Tang et al., 2024a). We then filter out meshes with more than 4,000 faces, yielding 39,232 meshes for training and 4,368 for testing. During training, we employ random scaling and random translation.

## 5 RESULT

### 5.1 TRAINING PHASE

Table 1 demonstrates the significant efficiency advantages of our approach during training on the ShapeNet dataset. The simplified interleaving architecture (**I+S**) reduces per-GPU memory consumption from 63GB to 51GB, a 20% reduction compared to the original interleaving architecture (**I**), while decreasing training time from 5 hours and 41 minutes to 5 hours and 24 minutes (5.0% improvement).

While the full attention model achieves relatively low peak memory usage (45GB/GPU), because Flash attention can achieve linear memory complexity even though the time complexity remains quadratic. However, it requires 30% longer training time than **I+S** (7 hours and 2 minutes vs. 5 hours and 24 minutes).

Our full architecture (**I+S+H**), which incorporates both simplified interleaving and the hourglass structure, delivers substantially greater improvements across all metrics. It reduces memory require-

ments to just 28GB per GPU (a 56% reduction compared to **I** and 38% compared to full attention) while completing training in only 3 hours and 47 minutes—a 30% reduction in training time versus **I+S** and 46% versus full attention. These dramatic efficiency gains make high-quality mesh generation more accessible even with limited computational resources.

Table 1: Resource usage comparison .

| Architecture | Memory | Epochs | Time |
|---|---|---|---|
| Full Attn | 45G×4 | 40 | 7h 02m |
| Linear Attn | 50G×4 | 40 | 5h 04m |
| **I** | 63G×4 | 40 | 5h 41m |
| **I+S** | 51G×4 | 40 | 5h 24m |
| **I+S+H** (Ours) | 28G×4 | 40 | 3h 47m |

Table 2: Inference performance comparison.

| Metric | Full Attn | Linear Attn | **I** | **I+S** | **I+S+H** (Ours) |
|---|---|---|---|---|---|
| Latency (ms/t) | 22.2 | 24.5 | 25.7 | 24.3 | **12.2** |
| Throughput (t/s) | 45.0 | 40.8 | 38.9 | 41.1 | **81.9** |
| GPU (GB) | 9.5 | **2.9** | 3.7 | 3.7 | 3.7 |
| KV Cache (GB) | 6.6 | **0.0** | 0.8 | 0.8 | 0.8 |
| Total Time (s) | 800.6 | 881.2 | 925.3 | 874.7 | **439.2** |

## 5.2 INFERENCE PHASE

This section analyzes inference performance across five model variants, with a batch size of 4 and 36000 tokens (4000 faces × 9 vertices). Standard interleaving (**I**) exhibits the highest latency (25.7 ms/token) due to SiLU activations and gate mechanisms, while our simplified version (**I+S**) demonstrates improved efficiency. Our complete architecture (**I+S+H**) achieves the lowest latency (12.2 ms/token) and fastest processing time (439.2 seconds), representing a 45% improvement over the most efficient baseline.

Regarding memory consumption, full attention models use significantly more memory, while linear attention models use the least. Our interleaving approaches consume similar amounts of memory, offering a trade-off between computational efficiency and model expressiveness.

## 5.3 COMPARISONS WITH OTHER METHODS

To validate the effectiveness of our proposed model, **iFlame**, we conducted a comprehensive quantitative comparison against leading unconditional mesh generation methods, **MeshGPT** and **MeshXL**. While several recent works, such as MeshAnything, EdgeRunner, and BPT focus on conditional generation and are thus not directly comparable, our evaluation ensures a fair assessment by using official checkpoints and identical training datasets where applicable. We excluded PivotMesh from our comparison as they did not provide checkpoints for the ShapeNet categories.

As shown in Table 3, iFlame demonstrates highly competitive performance despite significant advantages in model efficiency. Notably, MeshXL was pretrained on a massive 2.5 million samples before being fine-tuned on ShapeNet. In contrast, our model achieves these strong results with only **76M parameters**, compared to MeshGPT's 196.8M and MeshXL's 125M. Furthermore, iFlame requires substantially less training time—**less than 4 GPU days**, whereas MeshGPT requires 20 GPU days and MeshXL requires 80 GPU days. This highlights our model's superior efficiency in both parameter count and computational cost.

In the **Chair** category, iFlame achieves the highest **1-NNA score**, indicating superior sample quality, and secures the second-best **KID score**, rivaling the heavily trained MeshXL. For the **Table** category, our model outperforms both baselines in **Coverage (COV)**, demonstrating better distributional similarity to the ground truth data. It also achieves the best **1-NNA and KID scores**, underscoring its ability to generate high-fidelity and diverse meshes efficiently.

Table 3: Quantitative comparison with state-of-the-art mesh generation methods. We also show the number of parameters for each method. For evaluation, we randomly sampled 1,000 meshes from each model. ( best , 2nd best )

| Category | Methods | COV↑ | MMD↓ | 1-NNA | JSD↓ | FID↓ | KID↓ | Avg.# Faces |
|---|---|---|---|---|---|---|---|---|
| | MeshGPT (196.8M) | 81.79% | 26.33 | 57.03% | 26.0085 | 16.36 | 6.4 | 332.57 |
| Chair | MeshXL (125M) | 78.91% | 22.86 | 55.11% | 22.1043 | 8.76 | 1.3 | 416.18 |
| | **iFlame (76M)** | 77.64% | 26.98 | 54.79% | 25.0408 | 17.36 | 6.3 | 381.28 |
| | MeshGPT (196.8M) | 64.66% | 18.72 | 55.15% | 20.3536 | 9.81 | 2.9 | 238.63 |
| Table | MeshXL (125M) | 63.69% | 15.89 | 59.13% | 26.4926 | 8.94 | 2.8 | 328.81 |
| | **iFlame (76M)** | 65.44% | 17.73 | 55.15% | 25.5618 | 9.65 | 2.1 | 268.73 |

## 5.4 QUALITATIVE RESULTS

Figures 3 and 4 showcase the qualitative results of our proposed approach. As demonstrated in Figure 3, our model achieves mesh generation quality comparable to MeshGPT on ShapeNet while requiring significantly fewer parameters. For the more challenging Objaverse dataset, Figure 4 illustrates our model's capability to generate complex meshes with intricate geometric details and structures. Detailed generation quality metrics and ablation studies are provided in the **Appendix A.1 and A.2**.

## 5.5 NOVELTY ANALYSIS

We assess the novelty of our generated outputs in Fig. 5. In the figure, the green shapes represent our generative results, while the blue shapes illustrate their closest corresponding instances from the training dataset, as measured by Chamfer distance. The accompanying histogram shows the distribution of these minimum Chamfer distances for 1,000 generated chairs relative to their nearest training set neighbors. This analysis demonstrates that our approach can both faithfully reproduce learned patterns and explore creative possibilities within the design space.

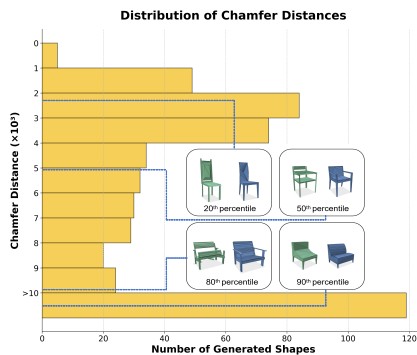

Figure 5: Novelty assessment of generated chair models.

## 5.6 SCALABILITY

We conducted an experiment on our model's scalability by varying the hidden state dimension across three configurations: 256, 512, and 1024. Each configuration was trained for 10 epochs under identical conditions to ensure fair comparison. Figure 6 illustrates the results of these experiments.

Our results show a clear relationship between model size and perplexity. Increasing the hidden state dimension consistently improves performance, with the 1024-dimensional model significantly outperforming the 256-dimensional version. This demonstrates that our interleaving attention mechanism effectively utilizes additional parameters without suffering from typical optimization challenges in larger models, suggesting potential for further improvements through additional scaling.

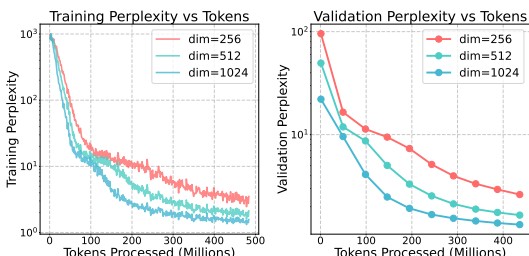

Figure 6: PPL comparison across different hidden state dimensions. Lower PPL indicates better performance.

## 6 CONCLUSION AND FUTURE WORK

We presented iFlame, an architecture for efficient mesh generation that balances computational efficiency with generative performance through interleaved full and linear attention mechanisms. Our approach demonstrates substantial efficiency improvements while maintaining high-quality output.

Future work includes scaling to larger meshes (10,000+ faces), increasing model capacity through larger embedding dimensions or deeper networks, exploring a mixture of expert architecture for parameter efficiency, and extending to conditional generation scenarios such as point-to-mesh or image-to-mesh tasks.

We believe iFlame represents a significant advancement toward making high-quality mesh generation more accessible and practical for real-world applications.

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
