

| Input | Full Attn | Linear Attn | I | I+S | I+S+H | Ground Truth |

Figure 1: Ablation study on ShapeNet.

# A  APPENDIX

## A.1  GENERATION QUALITY METRICS

Table 1 compares performance across different models and categories. For Token Accuracy, **I**, **I+S**, and **I+S+H** achieve 96.3%, matching **I** and only 0.2% below full attention, while linear attention achieves 95.2%. In Face Accuracy, **I+S+H** performs best among our proposed variants at 78.1%, nearly matching full attention's 78.3%, with **I+S** and **I** achieving 77.2% and 77.8% respectively. Linear attention lags significantly at 71.9%. For Perplexity, full attention, **I**, and **I+S** all achieve 1.14, with **I+S+H** slightly higher at 1.15 and linear attention at 1.18. These results demonstrate that our model maintains near-optimal performance while improving computational efficiency.

## A.2  ABLATION STUDY

We evaluate different model architectures by testing their ability to complete mesh geometry when given the first 50 faces as input (Fig. 1). The linear attention model produces the poorest results with disconnected meshes. Full attention generates more detailed outputs due to its greater capacity. The interleaving model (**I**) and (**I+S**) show intermediate performance levels. Our approach (**I+S+H**) achieves results comparable to full attention while maintaining computational efficiency, validating our architectural design choices.

## A.3  EFFICIENCY ANALYSIS

This section details iFlame's efficient token processing techniques, focusing on Selective Cache-Efficient Token Processing (Algorithm 2) and Interleaving Attention Processing (Algorithm 1). These algorithms enhance computational efficiency and memory usage during inference while preserving output quality. Through optimized caching and selective computation, iFlame handles longer sequences with limited resources. Additional generated results from the Objaverse dataset are presented in Figures 2, 3, and 4. Table 2 compares recent mesh generation approaches. Mesh sequence length remains a fundamental challenge, with researchers typically pursuing either: (1) autoencoder compression to represent faces with fewer tokens, or (2) sequence-compressing serialization methods. However, we focused on the architectural design for mesh generation. We can further incorporate other designs (*e.g.*, the sequence ordering (Chen et al., 2024b; Weng et al., 2024b)) to improve the performance.

| Category | Full attn | Linear attn | I | I+S | I+S+H |
|---|---|---|---|---|---|
| **Token Accuracy (%) ↑** | | | | | |
| Bench | 96.6 | 95.5 | 96.6 | 96.5 | 96.6 |
| Bookshelf | 98.1 | 97.2 | 98.1 | 97.8 | 98.2 |
| Chair | 94.9 | 93.4 | 94.8 | 94.7 | 94.7 |
| Display | 94.9 | 93.2 | 94.7 | 94.6 | 94.7 |
| Table | 97.4 | 96.3 | 97.3 | 97.2 | 97.3 |
| All | 96.5 | 95.2 | 96.3 | 96.3 | 96.3 |
| **Face Accuracy (%) ↑** | | | | | |
| Bench | 78.5 | 72.3 | 78.9 | 77.7 | 79.1 |
| Bookshelf | 87.6 | 81.9 | 87.3 | 86.0 | 87.5 |
| Chair | 70.4 | 63.7 | 70.0 | 69.6 | 70.3 |
| Display | 69.0 | 61.9 | 68.2 | 67.4 | 68.0 |
| Table | 83.2 | 77.1 | 82.5 | 82.0 | 82.9 |
| All | 78.3 | 71.9 | 77.8 | 77.2 | 78.1 |
| **Perplexity (PPL) ↓** | | | | | |
| Bench | 1.13 | 1.16 | 1.12 | 1.13 | 1.13 |
| Bookshelf | 1.07 | 1.09 | 1.07 | 1.08 | 1.07 |
| Chair | 1.22 | 1.26 | 1.22 | 1.22 | 1.23 |
| Display | 1.21 | 1.26 | 1.21 | 1.21 | 1.21 |
| Table | 1.10 | 1.13 | 1.10 | 1.10 | 1.10 |
| All | 1.14 | 1.18 | 1.14 | 1.14 | 1.15 |

Table 1: Accuracy comparison across different models and categories.

| Model | MeshGPT | MeshAnyth | MeshAnythv2 | MeshXL | PivotMesh | EdgeRunner | BPT | Meshtron | Ours |
|---|---|---|---|---|---|---|---|---|---|
| Unconditional Generation | ✓ | ✗ | ✗ | ✓ | ✓ | ✗ | ✗ | ✗ | ✓ |
| Attention | Full | Full | Full | Full | Full | Full | Full | Full | Interleaving |
| Networks | Plain | Plain | Plain | Plain | Plain | Plain | Plain | Hourglass | Hourglass |
| Core Contributions | Autoencoder | Scaling | Sequence Ordering | Scaling | Autoencoder | Autoencoder | Sequence Ordering | Architecture | Architecture |

Table 2: Comparison of different mesh generation models and their key characteristics.

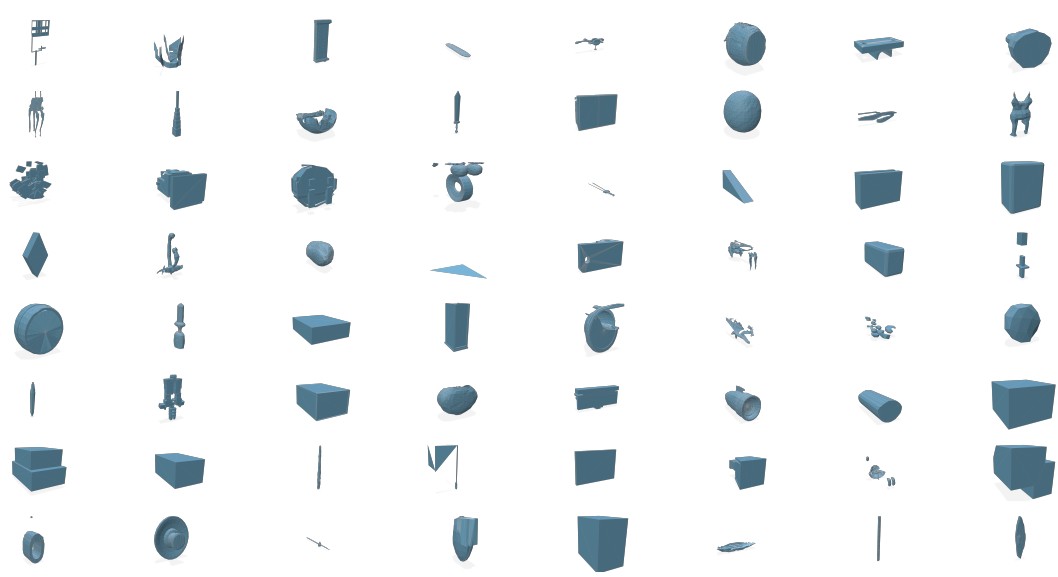

Figure 2: More generative results on Objaverse.

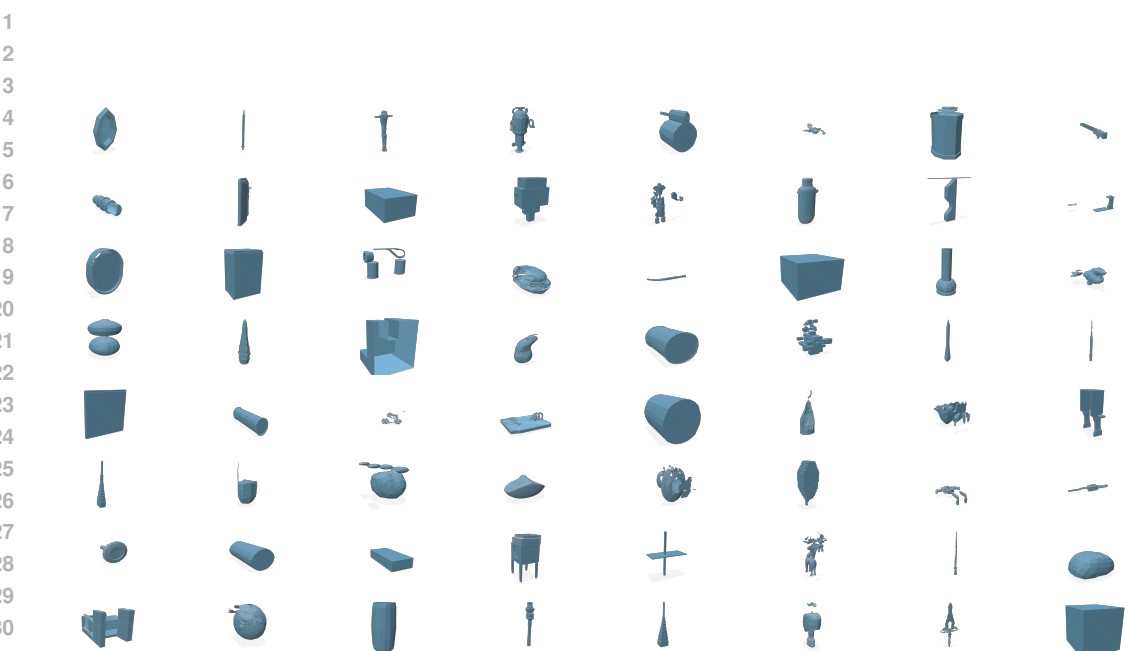

Figure 3: More generative results on Objaverse.

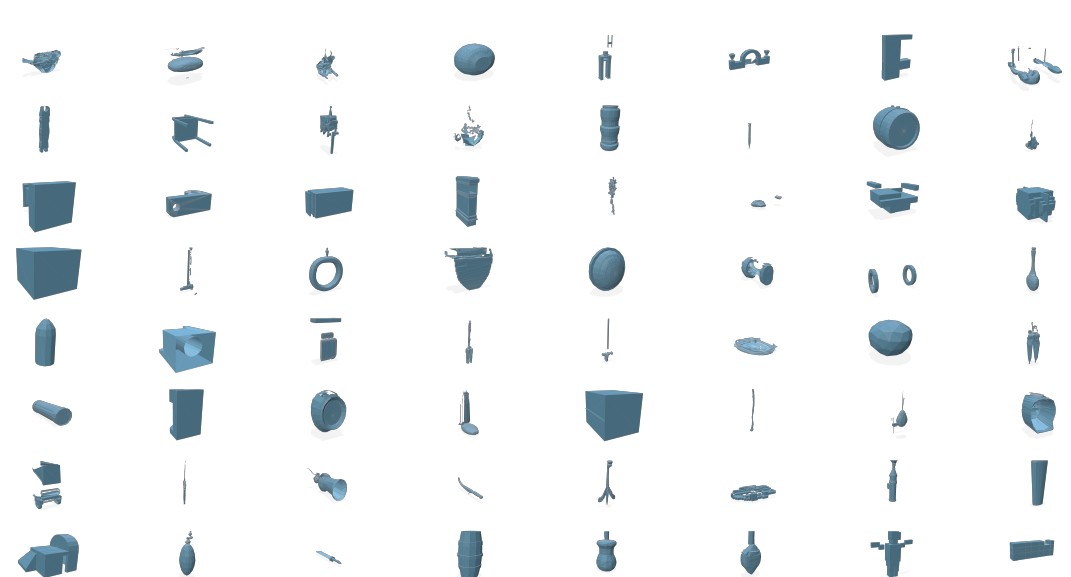

Figure 4: More generative results on Objaverse.

**Algorithm 1** Interleaving Attention Processing in iBlock with Selective Caching

**Require:** $\mathbf{x}_t$: Input embedding at position $t$
**Require:** $\mathcal{C}$: Key-value cache state
**Ensure:** $\mathbf{y}_t$: Processed representation
1: **function** INTERLEAVINGATTENTIONBLOCK($\mathbf{x}_t, t$)
2:     $\mathbf{x}'_t \leftarrow \text{LayerNorm}(\mathbf{x}_t)$                                                                $\triangleright$ Pre-normalization
3:     $\mathbf{q}_t, \mathbf{k}_t, \mathbf{v}_t \leftarrow \text{ProjectQKV}(\mathbf{x}'_t)$
4:     $\mathbf{q}_t, \mathbf{k}_t \leftarrow \text{ApplyRotaryEmbedding}(\mathbf{q}_t, \mathbf{k}_t, t)$                                   $\triangleright$ Position encoding
5:     **if** $t \bmod 4 = 0$ **then**                                                             $\triangleright$ Full attention
6:         $\mathcal{C}.\text{KV}[:, t \bmod L] \leftarrow (\mathbf{k}_t, \mathbf{v}_t)$
7:         $\mathbf{a}_t \leftarrow \text{FullAttention}(\mathbf{q}_t, \mathcal{C}.\text{KV})$
8:     **else**                                                                                 $\triangleright$ Linear attention
9:         $\mathcal{C}.\text{State} \leftarrow \text{UpdateState}(\mathcal{C}.\text{State}, \mathbf{k}_t, \mathbf{v}_t)$
10:         $\mathbf{a}_t \leftarrow \text{LinearAttention}(\mathbf{q}_t, \mathcal{C}.\text{State})$
11:     **end if**
12:     $\mathbf{z}_t \leftarrow \mathbf{x}_t + \mathbf{a}_t$                                                                     $\triangleright$ First residual connection
13:     $\mathbf{z}'_t \leftarrow \text{LayerNorm}(\mathbf{z}_t)$                                                             $\triangleright$ Second normalization
14:     $\mathbf{f}_t \leftarrow \text{FeedForward}(\mathbf{z}'_t)$                                                            $\triangleright$ Position-wise FFN
15:     $\mathbf{y}_t \leftarrow \mathbf{z}_t + \mathbf{f}_t$                                                                     $\triangleright$ Second residual connection
16:     **return** $\mathbf{y}_t$
17: **end function**

---

**Algorithm 2** Selective Cache-Efficient Token Processing in iFlame

---

**Require:** $x_t$: Input token at position $t$
**Require:** $\mathcal{S}$: Inference state with cached representations
**Ensure:** $y_t$: Output logits for current token
1: **function** PROCESSTOKEN($x_t$)
2:     $t \leftarrow \mathcal{S}.\text{position}$
3:     $x_t \leftarrow \text{Embed}(x_t)$                                              $\triangleright$ Token embedding and normalization
4:     **/\* Determine update schedule \*/**
5:     $\phi_0 \leftarrow$ True                                                       $\triangleright$ Update encoder stage 0
6:     $\phi_1 \leftarrow (t+1) \bmod 3 = 0$                         $\triangleright$ Update encoder stage 1
7:     $\phi_b \leftarrow (t+1) \bmod 9 = 0$                            $\triangleright$ Update bottleneck
8:     $\psi_0 \leftarrow (t+1) \bmod 3 = 0$                         $\triangleright$ Update decoder stage 0
9:     $\psi_1 \leftarrow$ True                                            $\triangleright$ Update decoder stage 1
10:    **/\* Encoder pathway (downsampling) \*/**
11:    **if** $\phi_0$ **then**
12:       $e_0 \leftarrow \text{EncoderBlock}_0(x_t)$
13:       $\mathcal{S}.\text{cache}_E[0][t \bmod 3] \leftarrow e_0$
14:    **end if**
15:    **if** $\phi_1$ **then**
16:       $z \leftarrow \text{Downsample}(\mathcal{S}.\text{cache}_E[0])$
17:       $e_1 \leftarrow \text{EncoderBlock}_1(z)$
18:       $\mathcal{S}.\text{cache}_E[1][\lfloor t/3 \rfloor \bmod 3] \leftarrow e_1$
19:    **end if**
20:    **/\* Bottleneck processing \*/**
21:    **if** $\phi_b$ **then**
22:       $z \leftarrow \text{Downsample}(\mathcal{S}.\text{cache}_E[1])$
23:       $b \leftarrow \text{BottleneckBlock}(z)$
24:       $\mathcal{S}.\text{cache}_D[0] \leftarrow \text{Upsample}(b)$
25:    **end if**
26:    **/\* Decoder pathway (upsampling) \*/**
27:    **if** $\psi_0$ **then**
28:       $d_0 \leftarrow \mathcal{S}.\text{cache}_D[0]$                          $\triangleright$ Retrieve upsampled state
29:       $d_0 \leftarrow d_0 + \mathcal{S}.\text{cache}_E[1]$                  $\triangleright$ Skip connection
30:       $d_0 \leftarrow \text{DecoderBlock}_0(d_0)$
31:       $\mathcal{S}.\text{cache}_D[1] \leftarrow \text{Upsample}(d_0)$
32:    **end if**
33:    **if** $\psi_1$ **then**
34:       $d_1 \leftarrow \mathcal{S}.\text{cache}_D[1]$                          $\triangleright$ Retrieve upsampled state
35:       $d_1 \leftarrow d_1 + \mathcal{S}.\text{cache}_E[0]$                  $\triangleright$ Skip connection
36:       $d_1 \leftarrow \text{DecoderBlock}_1(d_1)$
37:    **end if**
38:    $y_t \leftarrow \text{OutputProjection}(d_1)$                  $\triangleright$ Project to vocabulary
39:    $\mathcal{S}.\text{position} \leftarrow t+1$                    $\triangleright$ Update position counter
40:    **return** $y_t$
41: **end function**

---