# OpenReview forum: "iFlame: Interleaving Full and Linear Attention for Efficient Mesh Generation"
_ICLR.cc/2026/Conference — Submitted to ICLR 2026_

### Official Review · Reviewer_dKUP · 2025-10-25

**Soundness:** 2
**Presentation:** 3
**Contribution:** 3
**Rating:** 4
**Confidence:** 4

**Summary:**

This paper presents iFlame, a novel transformer architecture for efficient autoregressive mesh generation. The core idea is to address the trade-off between the computational cost of standard full attention, which scales quadratically with sequence length, and the representational limitations of linear attention, which struggles with long-range dependencies. The proposed solution is an interleaving mechanism that selectively combines both attention types, aiming to harness the efficiency of linear attention while maintaining the expressive power necessary for high-quality, scalable 3D data generation. The authors focus on demonstrating how this architectural change can lead to more efficient training for generative mesh models.

**Strengths:**

- The paper is well written and clearly motivated.
- The exploration on linear attention and more efficient training is important. It is pretty impressive to use this amount of compute to make mesh generation work.

**Weaknesses:**

The primary weakness is that the experiment scale is relatively small compared to related works. The model is only trained on 39K meshes, and the generalization capability is questionable since it is unconditional. While this is orthogonal to the main contribution of the work, it makes the scalability of the method less convincing.

**Questions:**

- Does the author believe that the linear attention could further scale up to larger models and datasets? What's the potential problems? Most recent mesh generation works still use a standard transformer, I'm curious about the authors' opinion on their choices.
- Have the authors tried performing conditional generation? It's hard to tell if the model can generalize to unseen objects with unconditional generation setting. Adopting a point cloud encoder shouldn't add lots of computation, so the proposed method should also show promising speed up in training.

---

> ### Author Response · Authors · 2025-11-21
>
> > **Q1: Scalability of Linear Attention to larger models/datasets.**
>
> **A1:** We strongly believe Linear Attention is scalable. The main current bottleneck is **infrastructural rather than theoretical**. While Flash Attention is highly optimized, standard libraries for efficient Linear Attention are still maturing. However, with recent community efforts (e.g., `flash-linear-attention`), kernel support is **rapidly improving**. As these libraries become standard, the scalability advantage of our linear-hybrid approach will become even more pronounced compared to full attention, especially for larger models and datasets.
>
> Crucially, this direction is being **validated by leading industry labs**. Recent large-scale explorations by the teams behind MiniMax [1], Kimi [2], and Qwen [3] have successfully demonstrated that linear attention mechanisms can be **scaled to massive parameter counts** with impressive performance. These industrial-grade results corroborate our premise: the $O(N)$ efficiency advantage of our linear-hybrid approach will become even more pronounced for large-scale mesh generation.
>
> *[1] MiniMax-01: Scaling Foundation Models with Lightning Attention*
>
> *[2] KIMI LINEAR: An expressive, efficient attention architecture*
>
> *[3] Qwen3-Next: Towards Ultimate Training & Inference Efficiency*
>
> > **Q2: Performance on Conditional Generation.**
>
> **A2:** To address your concern about generalization, we conducted a rigorous **Point Cloud-to-Mesh** generation experiment to compare iFlame against a standard Full-Attention baseline (aligned with MeshXL architecture).
>
> **Experimental Setup:**
> * **Task:** Conditional mesh generation from point clouds. We use a Shape2VecSet encoder to encode 8192 points (downsampled to 2048) and inject features via cross-attention.
>
> * **Dataset:** A subset of Objaverse containing $67328$ objects (<1000 faces). Both models are trained for 100 epochs.
>
> * **Baselines:**  We compare iFlame (Ours) against a Full-Attention Transformer (Baseline, equivalent to MeshXL/MeshGPT). Both use dim=512, heads=8, layers=24.
>
> * **Metrics:** Hausdorff Distance (HD) for worst-case deviation and Chamfer Distance (CD) for average proximity, evaluated on 100 unseen meshes.
>
>
> **Results:**
>
> | Method | Training Time (4x A100) | HD ($\downarrow$) | CD ($\downarrow$) |
> | :--- | :---: | :---: | :---: |
> | MeshXL (Baseline) | 86 hours | 0.0629 | 0.0324 |
> | **iFlame (Ours)** | **44 hours** | **0.0542** | **0.0296** |
>
> * **49% Training Speedup:** iFlame reduces training time from 86 hours to 44 hours.
> * **Superior Quality:** Despite faster training, iFlame outperforms the baseline in geometric quality, confirming robust generalization to conditional inputs.

---

### Official Review · Reviewer_1Ez7 · 2025-10-28

**Soundness:** 3
**Presentation:** 2
**Contribution:** 3
**Rating:** 6
**Confidence:** 3

**Summary:**

This paper proposes iFlame, an interleaved hourglass Transformer architecture that combines full attention with linear attention for efficient unconditional 3D mesh generation. By interleaving the two attention mechanisms, iFlame significantly improves training and inference efficiency while maintaining generation quality.  It also supports the generation of complex meshes with up to 4,000 faces, significantly outperforming existing methods.

**Strengths:**

Strong architectural innovation: This approach combines the staggered attention mechanism with an hourglass structure for mesh generation, effectively balancing efficiency and expressiveness.

Significant efficiency improvements: This approach significantly optimizes training time, memory usage, inference speed, and key-value caching.

Strong scalability: This approach successfully increases the maximum number of faces for unconditional mesh generation from 800 to 4,000, with potential for further expansion.

Comprehensive experimental design: This approach is compared with mainstream methods across multiple datasets and metrics, validating its effectiveness and versatility.

High practical value: This approach generates high-quality meshes after only two days of training on four A100 images, demonstrating promising prospects for practical deployment.

**Weaknesses:**

The baseline comparison is not comprehensive: It does not compare with recently proposed efficient Transformer variants (such as Mamba and RetNet).

Unexplored conditional generation capabilities: It focuses only on unconditional generation and has not verified its performance on conditional generation tasks (e.g., image/point cloud to mesh).

Generation quality still has room for improvement: Despite leading efficiency, it does not comprehensively surpass baselines such as MeshXL in some metrics (such as COV and MMD).

Lack of in-depth analysis of the architectural choice: Why was the "3 linear + 1 full attention" iBlock architecture chosen? Ablation experiments are lacking to support this.

Understanding the ability to model long sequences is not fully verified: While it supports 4,000 faces, its performance on longer sequences (e.g., 10,000+ faces) is not demonstrated.

**Questions:**

Comparison with Other Efficient Architectures:
Have you compared iFlame with other efficient sequence models (e.g., Mamba, RetNet, or Hyena) in terms of both efficiency and mesh quality?

Generalization to Conditional Generation:
Can iFlame be extended to conditional mesh generation tasks (e.g., image-to-mesh or point cloud-to-mesh)? If so, what adaptations would be necessary?

---

> ### Author Response · Authors · 2025-11-21
>
> > **Q1: Comparison with other efficient architectures (Mamba, RetNet).**
>
> **A1:** While SSM-based architectures like Mamba and RetNet are also forms of Linear Attention [1], our primary contribution is **pioneering the integration of efficient Linear Attention mechanisms into mesh generation**. We deliberately chose a simple Linear Attention as a **minimalist proof-of-concept**. This validates that linear attention is effective for mesh generation.
>
> Since Transformers are the dominant backbone (e.g., MeshGPT), our approach isolates the efficiency gains of linearization, establishing its foundational viability. Comparing against specific variants like Mamba is an exciting future direction, but not a prerequisite for this foundational contribution.
>
> *[1] KIMI LINEAR: An expressive, efficient attention architecture*
>
> > **Q2: Extension to Conditional Generation.**
>
> **A2:** Yes, iFlame is highly extensible to conditional tasks. We conducted a rigorous **Point Cloud-to-Mesh** generation experiment to compare iFlame against a standard Full-Attention baseline (aligned with MeshXL architecture).
>
> **Experimental Setup:**
> * **Task:** Conditional mesh generation from point clouds. We use a Shape2VecSet encoder to encode 8192 points (downsampled to 2048) and inject features via cross-attention.
>
> * **Dataset:** A subset of Objaverse containing $67328$ objects (<1000 faces). Both models are trained for 100 epochs.
>
> * **Baselines:**  We compare iFlame (Ours) against a Full-Attention Transformer (Baseline, equivalent to MeshXL/MeshGPT). Both use dim=512, heads=8, layers=24.
>
> * **Metrics:** Hausdorff Distance (HD) for worst-case deviation and Chamfer Distance (CD) for average proximity, evaluated on 100 unseen meshes.
>
>
> **Results:**
>
> | Method | Training Time (4x A100) | HD ($\downarrow$) | CD ($\downarrow$) |
> | :--- | :---: | :---: | :---: |
> | MeshXL (Baseline) | 86 hours | 0.0629 | 0.0324 |
> | **iFlame (Ours)** | **44 hours** | **0.0542** | **0.0296** |
>
> * **49% Training Speedup:** iFlame reduces training time from 86 hours to 44 hours.
> * **Superior Quality:** Despite faster training, iFlame outperforms the baseline in geometric quality, confirming robust generalization to conditional inputs.
>
> > **Q3: Architectural choice (3:1 ratio).**
>
> **A3:** The choice of the 3:1 ratio is structurally aligned with our Hourglass architecture and empirically verified.
>
> * **Architectural Alignment:** Our Hourglass encoder/decoder consists of scale blocks, where each block comprises exactly **4 layers**. A 3:1 ratio ensures consistent integration, allowing each scale block to conclude with a global synchronization (Full Attention) step. A 7:1 ratio would be structurally incompatible as it exceeds the block size.
> * **Empirical Verification:** We conducted ablation studies comparing Linear Attention, Full Attention, 1:1 hybrid, and 3:1 hybrid configurations:
>
> | Configuration | Token Accuracy | PPL | KV Cache (GB) |
> | :--- | :---: | :---: | :---: |
> | Full Attention | 96.5% | 1.14 | 6.6 |
> | Hybrid (1:1) | 96.4% | 1.14 | 3.3 |
> | **Hybrid (3:1)** | **96.3%** | **1.14** | **1.65** |
> | Linear Only | 95.3% | 1.18 | 0 |
>
> Increasing the attention density from our proposed 3:1 to 1:1 yields negligible gains (Accuracy +0.1%, PPL unchanged). However, adopting the 3:1 ratio significantly reduces the computational burden. Compared to the 1:1 configuration, **the 3:1 ratio further reduces the KV Cache by 50% (3.3GB to 1.65GB) without compromising perplexity.**
>
> The 3:1 ratio successfully retains the global modeling capabilities of Full Attention (matching PPL) while maximizing memory efficiency.
>
>
> > **Q4: Long sequence modeling.**
>
> **A4:** regarding long sequences, our Hybrid architecture demonstrates superior scalability compared to standard $O(N^2)$ Transformers. To substantiate this, we reference benchmarks comparing Linear Attention (Lightning Attention-2) against FlashAttention-2 [1].
>
> **Total Latency (Fwd+Bwd) Comparison**
>
> | Seq Len ($N$) | Lightning2 (Linear) | FlashAttn-2 | Speedup |
> | :--- | :---: | :---: | :---: |
> | 2,048 | 5.79 ms | 3.49 ms | 0.6x |
> | 4,096 | 11.55 ms | 11.63 ms | **1.0x (Crossover)** |
> | 8,192 | 23.03 ms | 42.38 ms | 1.8x |
> | 16,384 | 46.04 ms | 162.26 ms | **3.5x** |
> | 32,768 | 92.25 ms | 633.57 ms | **6.9x** |
>
> *[1] Source: https://github.com/OpenNLPLab/lightning-attention*
>
> As shown, a "crossover point" occurs at N=4096. Beyond this, the $O(N)$ complexity dominates. At $N=32K$, Linear Attention is nearly **7x faster**. Since iFlame utilizes Linear Attention for 75% of its layers, it effectively harnesses this long-context capability. Crucially,unlike standard Attention, where the KV cache grows linearly with sequence length, our Linear layers maintain a fixed-size state. This drastically reduces the KV cache footprint (e.g., 0.8GB vs. 6.6GB as shown in Table 1), uniquely positioning iFlame to generate extremely long sequences that would otherwise cause Out-Of-Memory (OOM) errors in standard Transformers.

---

### Official Review · Reviewer_WAMz · 2025-10-31

**Soundness:** 2
**Presentation:** 3
**Contribution:** 2
**Rating:** 4
**Confidence:** 4

**Summary:**

The paper proposes iFlame, an auto-regressive mesh generation method with hybrid linear - full attention. The hybrid attention optimizes the generation speed and memory cost, enabling both efficient training and sampling at a much lower cost than existing methods.

**Strengths:**

1. The paper proposes a hybrid architecture that can optimize training/inference speed while reducing the memory footprint of the KV cache.
2. The proposed method does not hurt much of the results.

**Weaknesses:**

1. The proposed method is merely replacing some of the full attention layers in the hourglass transformer. It would require further justification on why the architecture is designed this way, and why it is good for mesh generation.

2. As highlighted that linear attention "achieving lower performance compared to attention-based architectures" in line 58, it is important to validate the effectiveness of this observation on auto-regressive mesh generation.

3. What is face accuracy in Figure 1? I don’t think a truncated histogram is a good visualization method, as it can lead to misunderstandings about the multiplicative relationships between data points.

**Questions:**

See weakness part.

---

> ### Author Response · Authors · 2025-11-21
>
> > **Q1: Justification for the hybrid architecture.**
>
> **A1:** We respectfully argue that the design is not merely a replacement, but a strategic solution to the $O(N^2)$ bottleneck in long-sequence mesh generation. To the best of our knowledge, this work represents the **first exploration of Linear Attention mechanisms within the domain of mesh generation**.
>
> * **Motivation:** Pure Linear Attention is efficient ($O(N)$) but lacks the precision required for complex 3D topology. Full Attention is precise but computationally prohibitive for long sequences.
> * **Design Logic:** Our hybrid design effectively "recovers" global modeling capabilities by injecting Full Attention at specific intervals (the 3:1 design). This allows the model to maintain high geometric quality (comparable to Full Attention) while enjoying the linear complexity benefits for **75% of the layers**.
>
> > **Q2: Effectiveness of Linear Attention in auto-regressive mesh generation.**
>
> **A2:** We validated this explicitly by conducting ablation studies comparing Linear Attention, Full Attention, and our Hybrid configuration:
>
> | Configuration | Face Accuracy |
> | :--- | :---: |
> | Full Attention | 78.3% |
> | **iFlame (Ours)** | **78.1%** |
> | Linear | 71.9% |
>
> * **Pure Linear vs. Hybrid:** A model using purely linear attention suffers a significant drop in generation quality, with Face Accuracy decreasing by **6.4%**. This confirms our intuition that linear layers alone are insufficient.
> * **Our Solution:** In contrast, our iFlame approach sacrifices only **0.2%** in Face Accuracy compared to a computationally expensive Full-Attention model, while retaining the complexity benefits. This confirms that the hybrid design effectively mitigates the limitations of pure linear attention.
>
> > **Q3: Clarification on "Face Accuracy" and Figure 1 visualization.**
>
> **A3:**
> * **Metric Definition:** Face Accuracy is a strict structural metric derived from teacher-forcing accuracy. Since one mesh face is defined by **9 tokens**, a face is considered "accurate" **only if all 9 tokens are predicted correctly**. This makes it a rigorous measure of local geometric consistency.
> * **Visualization:** We respectfully clarify that **Figure 1** presents comparative bar charts across distinct performance metrics, not histograms representing data distributions. The visual scaling was deliberately designed to accommodate the significant differences between the baseline and our method.

---

### Official Review · Reviewer_Uv3N · 2025-10-31

**Soundness:** 3
**Presentation:** 3
**Contribution:** 2
**Rating:** 6
**Confidence:** 2

**Summary:**

This paper introduces iFlame, a Transformer-based architecture for high-resolution, unconditional 3D mesh generation. It tackles the trade-off between the expressiveness of full-attention and the efficiency of linear attention. The core idea is to design iBlocks, which interleave multiple linear attention layers with a single full attention layer to balance accuracy and cost.These iBlocks are further integrated into a multi-scale hourglass architecture that exploits the hierarchical structure of mesh data to shorten sequence lengths and enhance efficiency.

**Strengths:**

The combination of an interleaved attention mechanism within an hourglass structure is a conceptually novel design. It addresses the efficiency problem at both a block-level and global-level scale.
2 . Quantitative results demonstrate substantial gains in computational efficiency, training cost, and memory footprint, highlighting the method’s potential for practical deployment.

**Weaknesses:**

1 . The paper lacks direct numerical comparisons of inference latency and memory usage with external baselines (e.g.,MeshGPT, MeshXL). Including such results would strengthen claims about deployment efficiency.
2 . The fixed 3:1 ratio of linear to full attention layers is a key design choice, yet its empirical justification or sensitivity analysis is missing. Exploring alternative ratios (e.g., 1:1 or 7:1) would clarify its effect on the trade-off between performance and efficiency.

**Questions:**

1 . Could the authors further clarify the rationale for selecting the 3:1 ratio of linear to full attention layers within the iBlock? Was this choice guided by theoretical considerations, empirical tuning, or prior design heuristics?
2 . In Figure 5, the novelty analysis based on Chamfer distance is interesting. Could the authors provide qualitative examples to illustrate whether high-distance samples correspond to meaningful creative variations or to geometric artifacts?

---

> ### Author Response · Authors · 2025-11-21
>
> > **Q1: Comparison of inference latency/memory with external baselines (MeshGPT/MeshXL).**
>
> **A1:** Thank you for suggesting this comparison. We clarify that **Table 2** in our paper essentially serves as this direct comparison.
>
> **Performance Comparison**
>
> | Metric | Full Attention (MeshXL/MeshGPT) | **iFlame (Ours)** |
> | :--- | :---: | :---: |
> | **Throughput** (tokens/s) | 45.0 | **81.9** |
> | **GPU Memory** (GB) | 9.5 | **3.7** |
> | **KV Cache** (GB) | 6.6 | **0.8** |
>
> * **Equivalence of Baselines:** Methods like MeshXL and MeshGPT utilize the standard vanilla autoregressive Transformer architecture. Therefore, the "Full Attention" reported in Table 2 represents the theoretical and practical performance of the MeshXL/MeshGPT architecture.
> * **Performance Gap:** As shown above, **iFlame significantly outperforms this architecture**. We achieve a throughput of **81.9 tokens/s** compared to the baseline's 45.0 tokens/s.
> * **Memory Efficiency:** Practically, iFlame requires only **3.7 GB** of memory, whereas the MeshXL-equivalent baseline consumes 9.5 GB. This demonstrates substantial deployment advantages over current SOTA architectures.
>
> > **Q2: Rationale for the 3:1 ratio and sensitivity analysis.**
>
> **A2:** The choice of the 3:1 ratio is structurally aligned with our Hourglass architecture and empirically verified.
>
> * **Architectural Alignment:** Our Hourglass encoder/decoder consists of scale blocks, where each block comprises exactly **4 layers**. Setting a period of 4 (3 Linear + 1 Full) ensures consistent integration, allowing each scale block to conclude with a global synchronization (Full Attention) step. A 7:1 ratio would be structurally incompatible as it exceeds the block size.
> * **Empirical Verification:** We conducted ablation studies comparing Linear Attention, Full Attention, 1:1 hybrid, and 3:1 hybrid configurations:
>
> | Configuration | Token Accuracy | PPL | KV Cache (GB) |
> | :--- | :---: | :---: | :---: |
> | Full Attention | 96.5% | 1.14 | 6.6 |
> | Hybrid (1:1) | 96.4% | 1.14 | 3.3 |
> | **Hybrid (3:1)** | **96.3%** | **1.14** | **1.65** |
> | Linear Only | 95.3% | 1.18 | 0 |
>
> Increasing the attention density from our proposed 3:1 to 1:1 yields negligible gains (Accuracy +0.1%, PPL unchanged). However, adopting the 3:1 ratio significantly reduces the computational burden. Compared to the 1:1 configuration, **the 3:1 ratio further reduces the KV Cache by 50% (3.3GB to 1.65GB) without compromising perplexity.**
>
> The 3:1 ratio successfully retains the global modeling capabilities of Full Attention (matching PPL) while maximizing memory efficiency.
>
> > **Q3: Qualitative analysis of high Chamfer Distance.**
>
> **A3:** In the context of unconditional generation, Chamfer Distance (CD) measures the divergence between the generated sample and its nearest neighbor in the dataset. As illustrated in **Figure 3**, our method generates **plausible and creative mesh variations** that differ from the training data. This explains the higher CD compared to methods that produce fewer novel generations and tend to overfit the dataset.

---

### Meta-Review · Area_Chair_XrAi · 2026-01-07

**Summary:**

This paper receives mixed reviews of 2x marginal accepts and 2x marginal rejects. The strengths  are the arichitectural innovation and significant efficiency improvements with high practical values. The weaknesses are: 1) proposed method is merely replacing some of the full attention layers in the hourglass transformer; 2) experiment scale is relatively small compared to related works. The model is only trained on 39K meshes, and the generalization capability is questionable since it is unconditional; 3) Lack of in-depth analysis of the architectural choice. Despite the responses to the weaknesses in the rebuttal, the AC feels that the issues of small experimental scale and the lack of design novelty by merely replacing some of the full attention layers in the hourglass transformer remain. The AC thus follows the comments from the reviewers to reject the paper.

**Reviewer Concerns:**

The weaknesses are: 1) proposed method is merely replacing some of the full attention layers in the hourglass transformer; 2) experiment scale is relatively small compared to related works. The model is only trained on 39K meshes, and the generalization capability is questionable since it is unconditional; 3) Lack of in-depth analysis of the architectural choice. Despite the responses to the weaknesses in the rebuttal, the AC feels that the issues of small experimental scale and the lack of design novelty by merely replacing some of the full attention layers in the hourglass transformer remain.

**Reviewer Scores:**

From the rebuttal and lack of responses from the reviewers, it is unlikely that there will be a change in the scores. The final scores would remain at 2x marginal accepts and 2x marginal rejects.

---

### Decision · Program_Chairs · 2026-01-26

Reject